# COVID-19 and Virtual Nutrition: A Pilot Study of Integrating Digital Food Models for Interactive Portion Size Education

**DOI:** 10.3390/nu14163313

**Published:** 2022-08-12

**Authors:** Dang Khanh Ngan Ho, Yu-Chieh Lee, Wan-Chun Chiu, Yi-Ta Shen, Chih-Yuan Yao, Hung-Kuo Chu, Wei-Ta Chu, Nguyen Quoc Khanh Le, Hung Trong Nguyen, Hsiu-Yueh Su, Jung-Su Chang

**Affiliations:** 1School of Nutrition and Health Sciences, College of Nutrition, Taipei Medical University, Taipei 110, Taiwan; 2Smart Surgery Co., Ltd., Taipei 110, Taiwan; 3Department of Nutrition, Wan Fang Hospital, Taipei Medical University, Taipei 116, Taiwan; 4Department of Computer Science and Information Engineering, National Taiwan University of Science and Technology, Taipei 106, Taiwan; 5Department of Computer Science, National Tsing Hua University, Hsinchu 300, Taiwan; 6Department of Computer Science and Information Engineering, National Cheng Kung University, Tainan 701, Taiwan; 7Professional Master Program in Artificial Intelligence in Medicine, College of Medicine, Taipei Medical University, Taipei 110, Taiwan; 8Research Center for Artificial Intelligence in Medicine, Taipei Medical University, Taipei 110, Taiwan; 9Translational Imaging Research Center, Taipei Medical University Hospital, Taipei 110, Taiwan; 10Department of Adult Nutrition Counselling, National Institute of Nutrition, Hanoi 113000, Vietnam; 11Department of Clinical Nutrition and Dietetics, National Hospital of Endocrinology, Hanoi 12319, Vietnam; 12Department of Dietetics, Taipei Medical University Hospital, Taipei 110, Taiwan; 13Graduate Institute of Metabolism and Obesity Sciences, College of Nutrition, Taipei Medical University, Taipei 110, Taiwan; 14Nutrition Research Center, Taipei Medical University Hospital, Taipei 110, Taiwan; 15Chinese Taipei Society for the Study of Obesity (CTSSO), Taipei 110, Taiwan

**Keywords:** image-based dietary assessment, tele-dietetics, nutrition education, augmented, virtual reality, distance education, online learning

## Abstract

Background and aims: Digital food viewing is a vital skill for connecting dieticians to e-health. The aim of this study was to integrate a novel pedagogical framework that combines interactive three- (3-D) and two-dimensional (2-D) food models into a formal dietetic training course. The level of agreement between the digital food models (first semester) and the effectiveness of educational integration of digital food models during the school closure due to coronavirus disease 2019 (COVID-19) (second semester) were evaluated. Method: In total, 65 second-year undergraduate dietetic students were enrolled in a nutritional practicum course at the School of Nutrition and Health Sciences, Taipei Medical University (Taipei, Taiwan). A 3-D food model was created using Agisoft Metashape. Students’ digital food viewing skills and receptiveness towards integrating digital food models were evaluated. Results: In the first semester, no statistical differences were observed between 2-D and 3-D food viewing skills in food identification (2-D: 89% vs. 3-D: 85%) and quantification (within ±10% difference in total calories) (2-D: 19.4% vs. 3-D: 19.3%). A Spearman correlation analysis showed moderate to strong correlations of estimated total calories (0.69~0.93; all *p* values < 0.05) between the 3-D and 2-D models. Further analysis showed that students who struggled to master both 2-D and 3-D food viewing skills had lower estimation accuracies than those who did not (equal performers: 28% vs. unequal performers:16%, *p* = 0.041), and interactive 3-D models may help them perform better than 2-D models. In the second semester, the digital food viewing skills significantly improved (food identification: 91.5% and quantification: 42.9%) even for those students who struggled to perform digital food viewing skills equally in the first semester (equal performers: 44% vs. unequal performers: 40%). Conclusion: Although repeated training greatly enhanced students’ digital food viewing skills, a tailored training program may be needed to master 2-D and 3-D digital food viewing skills. Future study is needed to evaluate the effectiveness of digital food models for future “eHealth” care.

## 1. Introduction

Quantifying food portion sizes is a core skill of dieticians, and it is traditionally taught through training in a kitchen-based laboratory. Coronavirus disease 2019 (COVID-19) resulted in a sudden shift from the classroom to virtual learning, whereby teaching was conducted remotely and on digital platforms. With the sudden shift away from kitchen-based training, digital food aids and a new paradigm for virtual food portion size education were urgently needed.

Food photography, as a portion size estimation aid (PSEA), has long been used to assist traditional dietary assessment methods [1]. PSEAs can be classified as two-dimensional (2-D) (e.g., drawings, food atlas, and digital 2-D images) [2,3,4,5] and three-dimensional (3-D) aids (e.g., food models such as balls and plastic food models) [2,6]. The advantages of 2-D food images are their low cost, easy accessibility, and ability to depict numerous portion sizes [1]. The development of smartphones has promoted the use of image-based records as an independent dietary assessment tool. Studies suggested that 2-D image-based dietary assessments (IBDAs) provide a convenient way of self-reporting food intake [7,8,9] and are as valid as traditional dietary assessment methods such as 24-h dietary recall and weighted food records [10]. Our previous work showed that the integration of 2-D digital food images into a formal dietetic training program increased dietetic students’ digital dietary assessment skills [11]. Although Taiwanese students’ [11] digital food viewing skills were comparable to those of American/Australian dietetic students [12] and Malaysian dieticians [13], the accuracy of food portion size quantification remained low [11].

While food estimates based on 2-D images are often preferred in practice, significant problems exist. 2-D images do not provide complete spatial information and lack depth perception. They rely heavily on perceptions and abstract conceptualization skills [14]; thus, it is difficult to perceive and conceptualize food portion sizes based on 2-D images. In contrast, 3-D food models are thought to require less cognitive effort and provide better volume conceptualization of food portions [6]. However, earlier findings suggested that conventional 3-D PSEAs (e.g., plastic food models) were not very suitable for studies due to their limited food types and portion sizes [1]. Virtual and augmented reality (mixed augmented reality, MAR), a new era in the field of nutrition with the presentation of food dimensions and volumes in a virtual interactive 3-D model, might help overcome the above-described problems with 2-D images [15]. 3-D photogrammetric software has rapidly advanced and now offers affordable and accessible means of creating interactive digital 3-D food models from several 2-D images [16]. Having vivid food models which look similar to real food may enable better conceptualization of food portions [15]. Only a few studies have so far documented the effectiveness of MAR technology in nutrition-related work, mainly in changing eating behaviors [17], improving nutrition knowledge [18], assisting in portion estimations [19], and presenting standard portion servings [20]. Rollo et al. [20] recently showed the effectiveness of ServAR, a MAR application that displays a virtual standard food serving on a plate as a way to help control portion sizes when serving food [20].

Dietetic students must learn multiple practical skills involved in dietary assessments such as culinary skills, food classification, and food portion size estimation. Usually, these contents are taught through face-to-face practical sessions in which real food is used. The global irruption of COVID-19 at the beginning of 2020 resulted in a sudden shift from the classroom to online learning, whereby teaching is undertaken remotely and on digital platforms [21]. A virtual classroom that replaces traditional practical nutrition teaching was required during the pandemic. This has made it difficult for practical nutrition courses, which are typically kitchen-based laboratory courses. In this regard, teachers must encourage the use of new technologies to ensure that students continue to receive the best possible training. The rise of MAR technology offers a promising tool for teaching food portion size digital food models. Digital food models, both 2-D and interactive 3-D food models, can be shared online, on public web pages, and can be embedded in virtual learning environments and online courses [16]. Currently, the effectiveness of 3-D and MAR technologies in food portion size education remains unclear. To our knowledge, there is also no formalized nutrition education program that incorporates digital food viewing skills into a dietetic training program. As a pilot attempt, we integrated interactive 3-D and 2-D digital food models, as virtual nutrition education tools, into dietetic curricula and evaluated students’ digital food viewing skills over nine months of repeated training. The aim of this pilot study was to implement digital food models (2-D food images and interactive 3-D digital models) into the formal dietetic training program and evaluate the effectiveness of virtual food portion size education. We hypothesized that the incorporation of digital food models may improve the accuracy of food identification and portion size quantification. The specific aims were: (1) to compare and explore the level of agreement between two digital food models (2-D and 3-D) before the outbreak of COVID-19 (first semester); (2) to evaluate the effectiveness of virtual food portion size education using digital food models during the COVID-19 outbreak (second semester), and (3) to evaluate students’ receptiveness and response to the educational integration of digital food models.

## 2. Materials and Methods

### 2.1. Participants

Because our pedagogical framework was developed and tailored for the Nutritional Practicum (NP) course, the inclusion criterion was being a second-year student enrolled in the NP course at the School of Nutrition and Health Sciences, Taipei Medical University (TMU) (Taipei, Taiwan) between September 2019 and July 2020. All enrolled students were informed about the research purposes in advance via email, and none declined to participate in this study. The study protocol was approved by the Institutional Review Board of TMU (N201904035).

### 2.2. Study Design

#### 2.2.1. Integration of Digital Food Models into a Formal Dietetic Training Program

Our earlier work [11] described a pedagogical framework for integrating 2-D image-based dietary assessments into the NP course. As a pilot attempt to create digital food models to assist virtual food portion education during the COVID-19 outbreak, we further developed interactive 3-D food models and integrated digital food models (2-D and 3-D) into the NP course. Figure 1 shows the overall flowchart of the educational integration of digital food models into the NP course.

We used a digital scanner stand (80 cm (height) × 80 cm (width) × 80 cm (length)) (DEEP LED Photo Accessories, Guangdong, China) to take 360° videos and 2-D food images. Foods (single foods or complex food sets) were placed on a standardized plate (small: 17 × 17 cm; large: 26 × 26 cm) or in a standardized bowl (small: 13 × 7.5 cm; medium: 16.5 × 7.5 cm; large: 22 × 8 cm) and drinks were placed in a standardized cup (width 10 cm × height 16 cm). Plates or bowls were then placed on a marked region (20 × 20 cm) on the scanner stand with fiducial markers (8 × 5 cm) as a reference object for the interpretation of the color and size of the food [12,22]. The 360° videos were taken with a video recorder mounted on a tripod with the lens 40 cm away from the horizontal plane of the food. Three video clips were filmed at angles of 30°, 45°, and 60° while self-rotating 360° in the vertical plane for 20 s for each food. Moreover, 2-D food images were taken using an AngleCam© application (Derekr Corp., Google Play). One image was taken at a camera angle of approximately 90° and a distance of 50 cm above the table mat. A second photograph was taken with an approximately 45° camera angle and 50 cm in height [11].

To create 3-D food models, the 360° videos were imported into Agisoft Metashape Professional Edition (vers. 1.6.2, Agisoft, St. Petersburg, Russia) to generate a dense point cloud, 3-D meshes, and texture mesh as shown in Figure 2. Agisoft Metashape is stand-alone software that processes 3-D photogrammetrics of digital videos and generates 3-D spatial data. The 3-D food models were then uploaded to the Sketchfab website https://sketchfab.com/susanlab108/collections (accessed on 1 September 2019), (Sketchfab, New York, NY, USA). To test the effectiveness of virtual food portion size education, we embedded digital food models in LimeSurvey vers 3.23.7, (LimeSurvey, Hamburg, Germany), which supports both 2-D and 3-D image display, and students could resize or rotate the 3-D food models via the platform.

#### 2.2.2. Evaluation of Digital Food Viewing Skills

##### First Semester: Comparison between 2-D and Interactive 3-D Food Models

In the first semester, which was conducted in a kitchen-based laboratory before disruption due to COVID-19, we trained students on how to take standardized food images in the classroom. A hands-on 3-D workshop was held to teach students how to create 3-D food models. The goal was to compare and explore the level of agreement among students’ digital food viewing skills when applying the two (2-D and 3-D) digital food models. In the exams, five food sets (red bean dorayaki, wonton noodles, roasted chicken, sweet corn, and sweet potato) with 12 food items were prepared, and the actual weight of each food set was recorded with a standardized scale by trained nutritionists. A nutritionist calculated portion sizes (as exchanges of Taiwan standard servings-Ex) and total calories according to the actual weight of each food set. Standardized 2-D food images (at 45° and 90°) and a virtual interactive 3-D model of those food sets were constructed prior to the exam. In the exam, students were asked to report all food ingredients and quantify the ground truth weight, Ex, and total calories of each set. A questionnaire was used to understand students’ receptiveness and responses to the integration of 2-D and 3-D food models into the NP course. Food image viewing times were restricted to 5 min per image.

##### Second Semester: Virtual Food Portion Education Using Combined (2-D and 3-D) Digital Food Models

When COVID-19 hit Taiwan in early January 2020, face-to-face teaching was immediately canceled in order to promote social distancing. The NP course was switched to online teaching. An online platform was established which comprised standardized 2-D images (45° and 90°) and interactive 3-D food models that covered a wide variety of foods and portion sizes. Students could resize or rotate the 3-D food models using a rotation button on the website to assess food dimensions and volumes. Students could practice digital food viewing skills outside of classes throughout the entire semester via the Sketchfab website. Eleven food sets of the most commonly consumed foods (sweet potato, beef noodles, fried rice, chicken bento, fish bento, hamburger, bun, dumplings, tempura, fried oyster egg, and stinky tofu) in Taiwan were selected for the final exam evaluation. In the exam, a pair of standardized 2-D images and a corresponding virtual 3-D model were provided for each meal set. Students were asked to report (1) the names of the food items, (2) food portion sizes (weight and exchange), and (3) the total calories (kcal) of each meal set.

##### Definition of Food Identification and Quantification Accuracy

Food ingredient identification was categorized into “accurate”, “inaccurate”, and “omitted”. To calculate the accuracy of food quantification, the food weight and calories estimated by students were compared to the ground truth weight and calories. We evaluated students’ food volume estimations as a food portion size quantification accuracy and percentage estimate errors. Details of the definition and calculation of the accuracy of food identification and quantification were presented in our previous study [11].

### 2.3. Data Analysis

GraphPad Prism 8 (GraphPad Software, La Jolla, CA, USA) and SPSS software were used for all analyses (vers. 23.0, IBM, Armonk, NY, USA). To evaluate if the data were normally distributed, the Kolmogorov–Smirnov test was used. Nonparametric data are reported as the mean and 95% confidence interval (CI) for normally distributed data or the median and interquartile range (IQR) [quartile 1 (Q1); quartile 3 (Q3)] for normally distributed data. The correlation between students’ performance for the 2-D and 3-D food models was evaluated using Spearman’s coefficient; coefficients of >0.5 and >0.7 indicated moderate and high degrees of correlation, respectively.

Individual differences between 2-D and 3-D estimates for each food item were calculated using the following Equation:$$\mathrm{Difference}\;(2-D-3-D)\%=\frac{\mathrm{student}’s\;3-D\mathrm{estimate}\;-\;\mathrm{student}’s\;2-D\mathrm{estimate}\;}{\mathrm{student}’s\;2-D\mathrm{estimate}}\;\%.$$

In the first semester, we compared students’ digital food viewing skills when using the 2-D and 3-D food models. For this purpose, students were separated into “equal vs. unequal performers” based on their 2-D and 3-D performances of portion size estimations. Equal and unequal performers were defined as respective performance differences between 2-D and 3-D estimates of <6 or ≥6 out of 12 tested food items in the first semester. In the second semester, the effectiveness of the educational integration of digital food models (both 2-D and 3-D) into the NP course was determined by the food quantification accuracy and median percentage error. A pairwise Chi-square test was used to compare percentages of food quantification accuracy. The Wilcoxon rank-paired test was used to compare an individual’s estimated percentage error between 2-D and 3-D, and the Mann–Whitney test was used to compare estimated percentage errors between the two groups of students (equal vs. unequal performers). An individual’s distribution of estimation errors was visually explored using a scatterplot. The significance level was set to *p* < 0.05.

## 3. Results

In total, 65 students completed the first and second semester exams. Participants were primarily identified as female (77%), second-year undergraduate students (aged 20~22 years) in the School of Nutrition and Health Sciences (95%), or with a double major (5%) in Nutrition and Public Health, Food Safety, Nursing, or Gerontology Health Management.

### 3.1. Comparison between 2-D and Interactive 3-D Digital Food Viewing Skills

In the first semester, the aim was to compare a student’s ability to perform digital food viewing skills when using interactive 3-D food models and when using 2-D food images (Table 1). Table 1 shows no statistical difference between the 2-D and 3-D food viewing skills of food identification (2-D: 89% vs. 3-D: 85%) and quantification (within ±10% difference of total calorie) (2-D: 19.4% vs. 3-D: 19.3%). The Spearman correlation analysis showed moderate to strong correlations of estimated total calories (0.69~0.93; all *p* < 0.05) between the 3-D and 2-D models across all food items. The omission rate was highest for cooking oil/butter (3-D: 46.3% vs. 2-D: 46.3%) and sauces/condiments/dressings (3-D: 32.8% vs. 2-D: 32.8%.

We next stratified students into “equal” (*n* = 45) and “unequal” performers (*n* = 20) based on their portion size estimation accuracy using the 2-D and 3-D models. Figure 3A shows that equal performers achieved higher food quantification accuracies than did unequal performers (equal performers: 28% vs. unequal performers: 16%, *p* = 0.041). Unequal performers also had higher median percentage errors in estimating amorphous food compared to equal performers [equal performers: median 6% (IQR: −2.75%, 23.25%) vs. unequal performers: median 14.75% (IQR: −2.75%, 33.38%), *p* < 0.05] (Figure 3B). For those students who struggled to perform both 2-D and 3-D food viewing skills, interactive 3-D food models may help them more than 2-D models (within ±10% difference of total calorie estimation: 2-D: 14% vs. 3-D: 19%) (C), especially when estimating food portions of vegetables (2-D: 4% vs. 3-D: 20%, *p* < 0.001) and condiments (2-D: 6% vs. 3-D: 17%, *p* < 0.05) (D) (Figure 3).

### 3.2. Student’s Receptiveness towards Educational Integration of Digital Food Models

Table 2 summarizes students’ receptiveness and responses toward integrating 2-D and 3-D digital food models into the NP course. One-third (31%) of students thought that real food visualization was easier for them to perform food identification, followed by no difference between the 2-D and 3-D models (28%), and 2-D images alone (23%). For food quantification, 38% of students thought that real food visualization was easier, followed by the combination of 2-D and 3-D food models (26%). The majority of students (71%) thought that digital food models (both 2-D and 3-D) would be helpful in conducting virtual dietary assessments (71%), and 82% of students supported the educational integration of digital food models into the formal NP course (Table 2).

### 3.3. The Effectiveness of Virtual Food Portion Size Education Using Digital Food Models

Due to the rising number of COVID-19 cases and the school shutting its doors in the second semester, virtual food portion education was conducted with the aid of 2-D and interactive 3-D food models as teaching tools. Table 3 shows that overall accuracies of digital food viewing skills of food identification and portion size estimation improved to 91.5% and 42.9%, respectively. Students who struggled to master both 2-D and 3-D digital food viewing skills (unequal performers) in the first semester also improved to levels comparable to those of students who did not struggle (equal performers: 44% vs. unequal performers: 40%) (Figure 4A). At 9 months, repeated training in digital food viewing skills also revealed a greater improvement in the estimation accuracy of amorphous foods, fillings/stuffing, and condiments (all *p* < 0.0001), but not vegetables (B), although the estimated error was substantially reduced compared to the first semester (C) (*p* < 0.05) (Figure 4).

## 4. Discussion

Food portion size estimation is a critical yet challenging task in dietary assessments. The present pilot study developed a novel pedagogical framework in which a virtual interactive food portion education paradigm was integrated into the formal dietetic curriculum. We found that nearly half (42%) of junior undergraduate students were able to estimate calories within ±10% after 9 months of repeated training. The use of interactive digital food models (both 2-D and 3-D) (42%) yielded a greater estimation accuracy than 2-D alone compared to our previous study (32%) [11], to Australian and American dietetic students (38%) [12], and to nutrition professionals in Malaysia (24~32%) [13]. The majority of students thought that digital food models (both 2-D and 3-D) were useful tools for “virtual dietary assessments” (71%) and that the educational paradigm of interactive food portion education should be formally integrated into the dietetic curriculum. Importantly, we also observed that students’ digital food viewing skills may have been affected by the type of digital food model. While most students appeared to show equal comprehension between the 2-D and 3-D models (equal performers), some students comprehended the digital food models unequally. For those students who struggled to assess digital food models (unequal performers), interactive 3-D food models might help them better than 2-D models. However, constructing high-quality 3-D food models using Agisoft Metashape is time-consuming and labor-intensive. Food viewing skills also differed between the 2-D and 3-D models, and students may need a tailored training program to master digital food viewing skills using 2-D and 3-D models.

After a 9-month repeated training of digital food viewing skills, we observed substantial improvements in food identification (91.5%) and the estimation accuracies of amorphous foods (58%), fillings (52%), and condiments (38%), and to a lesser extent, vegetables (6%). Repeated training is known to reduce measurement errors and improve the accuracy of food volume estimations [11,13,23]. The overall performance in the current study was better than previous studies using only 2-D images [11,13,23]. This suggests that the use of combined digital food models (2-D and 3-D) may have an advantage over 2-D images alone. 3-D photogrammetry offered a rotatable multi-view, with functionality that enabled enlarging the image and interacting with the user, which may have aided students in conceptualizing the food volume. The zoom-in operation enlarges the size of the image, which may facilitate users in identifying small food ingredients (e.g., fillings and stuffing) or condiments (cooking oil, sugars, and sauces). To our surprise, the incorporation of 3-D food models did not improve the estimation accuracy of vegetables (lettuce, salads, and cabbage). Studies suggested that these errors might not be limited to study participants or the type of approach. Dietitians have trouble quantifying vegetables using both real foods [24,25] and 2-D food images [13,26]. Vegetables are made up of several parts that are stacked on top of each another. Thus, even 3-D models that are zoomable and rotatable to provide an all-around perspective, cannot enhance the food portion accuracy if overlapping portions of a food cannot be seen. Another difficulty in quantifying vegetables is that the volume of cooked vegetables may change significantly from raw vegetables. As Asians tend to eat cooked vegetables, students normally learn the effect of cooking processes on food volumes through cooking, but virtual classes cannot provide such training.

While 2-D food images have been widely used as an independent dietary assessment [10,27], few studies have investigated the use of interactive 3-D food models for portion size education and virtual dietary assessments. We hypothesized that interactive 3-D food models may have an advantage over 2-D because this modality can (1) enhance the interactivity between students and digital food models, especially with remote learning, and (2) provide better volume conceptualization of food portion sizes through its zoomable and rotatable function. We found that the 3-D performance of dietetic students was moderately to strongly correlated with their 2-D performance, and for most students, there were no estimation differences between the 2-D and 3-D models. This finding contrasts with earlier work by Chung et al. in which the estimation accuracy of food volumes was higher with 2-D images compared to 3-D images, which were presented as a 360° self-rotating video clip [28]. It should be noted that in the first semester, students were unfamiliar with the interactive 3-D food models, and using both digital food models at the same time may have been confusing and irritating for novice users. Students’ receptiveness and responses to the educational integration of digital food models were: “The advantage of 3-D is it provides better volume estimations than 2-D; however, 2-D has better image quality than 3-D and is better for ingredient identification”. “It is easier to master the digital food viewing skills using 2-D than 3-D”. “I found that the 3-D model was more difficult for volume estimation because food portions changed after I performed the zoom-in function and enlarged the image”. Overall, this suggests that although 3-D allows users to see the food volume better than 2-D, it may also cause portion distortion if users do not know how to operate it. To address these comments, we trained students in how to prevent portion distortion through the use of fiducial markers that were present in the 3-D model. The inconsistent quality of 3-D food images may also have affected students’ digital food viewing skills. Although several technical-related issues need to be addressed, the majority of students thought that 3-D models have advantages over 2-D models, and digital food viewing skills are core skills for future tele-dietetics.

Several factors may have affected the quality of the 3-D photogrammetry, including a short adaption time, the resolution and number of photographs, the texture and shape of the food and its containers, and the specifications of the computer hardware [16]. Curricular changes typically take years to investigate, implement, and evaluate. However, the COVID-19 pandemic compelled us to make a drastic shift to online teaching in a short space of time, and we only had a few weeks to adapt to the new 3-D photogrammetric technology. In our experience, 3-D photogrammetric software struggles with translucent, white, and textureless foods such as rice, noodles, and drinks. Surfaces of food containers that are shiny, mirrored, or transparent also cause trouble when constructing 3-D food models [29], which may have resulted in a failure of food model construction. Moreover, complex foods containing sauces or small food ingredients may result in less robust and realistic 3-D photogrammetric models. Time consumption was also an issue at that time as we lacked manpower and computer resources. The photogrammetric approach was found to take variable amounts of time to photograph specimens, as it depends on the model, orientations needed, and angles used [16]. In our case, the post-processing time in photogrammetry was extremely variable. It took 2~8 h to construct one model due to the model complexity and performance of the computer.

Food image viewing is a vital skill that connects dieticians to e-health. One of the greatest advantages of conducting food portion size training via 2-D and 3-D models is that students can access to the contents from anywhere and can move or magnify the models. Our framework can be used to replace the traditional face-to-face dietetic training and boost students’ motivation and adherence to virtual e-learning in the era of COVID-19. Additionally, the MAR-3-D environment allows students to view and compare numerous models simultaneously, which is more challenging to do with 2-D images. Virtual and augmented reality, however, feature cutting-edge technology that can demonstrate realistic food 3-D models and concepts to students. Better 3-D scanners, photogrammetry software, and algorithms can increase the quality of 3-D models, but the level of detail of food that can be represented in the MAR viewpoint is limited. Our study is the first step in the establishment of a more complete research program for evaluating dietetic training curricula and the future development of integrating new technology into portion size education. Therefore, we propose to utilize digital food models that allow students to observe and manipulate them from VR and AR viewpoints in the following study.

### Strengths and Limitations

Strengths of this pilot study include a high completion rate (100%), longitudinal repeated training over a 9-month period, and a reasonable number of dietetic students (*n* = 65) who participated in the novel pedagogical framework of a virtual food portion size educational program. We also included a wide range of the most commonly consumed Taiwanese foods and food portion sizes during the courses. Our 2-D and 3-D food models represent actual habitual food intake and not only standard portion sizes. The main limitation of this study is the inconsistency of the 3-D quality due to its complicated development and configuration of the new technology, as discussed in the previous section. Another limitation is the participant sampling. As the study used second-year dietetic students on the NP course as the target learners, we could not provide the sample size calculation and power analysis in the context of this study. Limited numbers of food items and participants also highlight the need for future collaborations in order to recruit a large-scale representative population and confirm the validity and effectiveness of the new digital food portion size educational paradigm.

## 5. Conclusions

Shifting to virtual teaching in the time of the COVID-19 pandemic has pushed the digitalization of educational systems. The present pilot study proposed a novel pedagogical framework for the educational integration of digital food models into a dietetic course. Although repeated training greatly enhanced students’ digital food viewing skills, a tailored training program may be needed to master digital food viewing skills using 2-D and 3-D models. Our findings indicate that immersive MAR technology may be suitable for training dietetic students, especially in portion size estimation skills. We believe that our experience in providing the framework will lead to broader use of this technology and, potentially, the emergence of newer applications and innovations in dietetic training.

## Figures and Tables

**Figure 1 nutrients-14-03313-f001:**
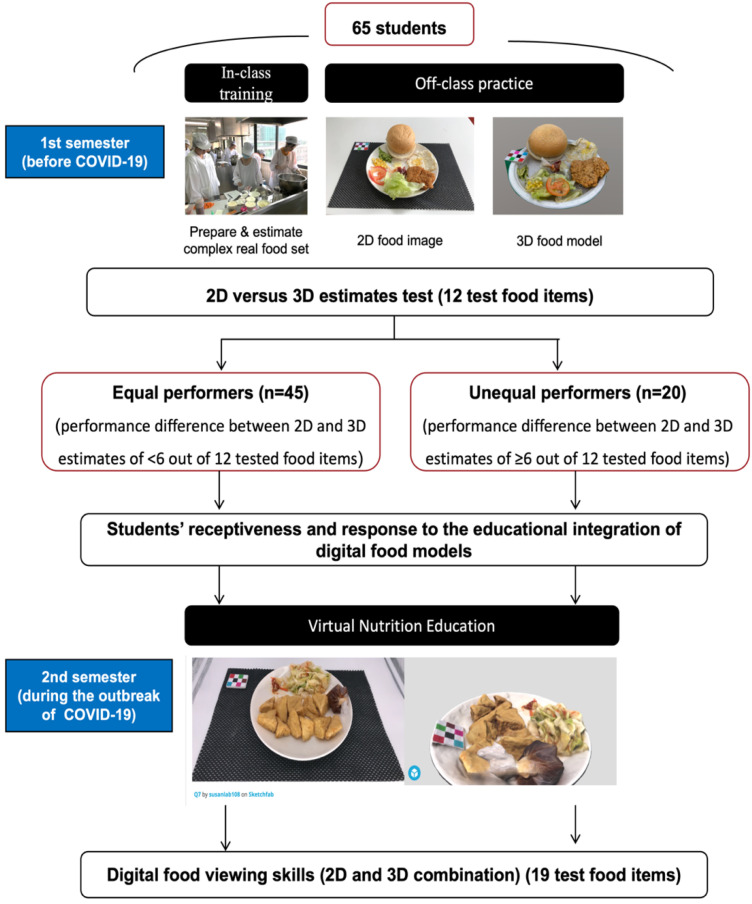
Study flowchart of educational integration of digital food models into a nutritional practicum course.

**Figure 2 nutrients-14-03313-f002:**
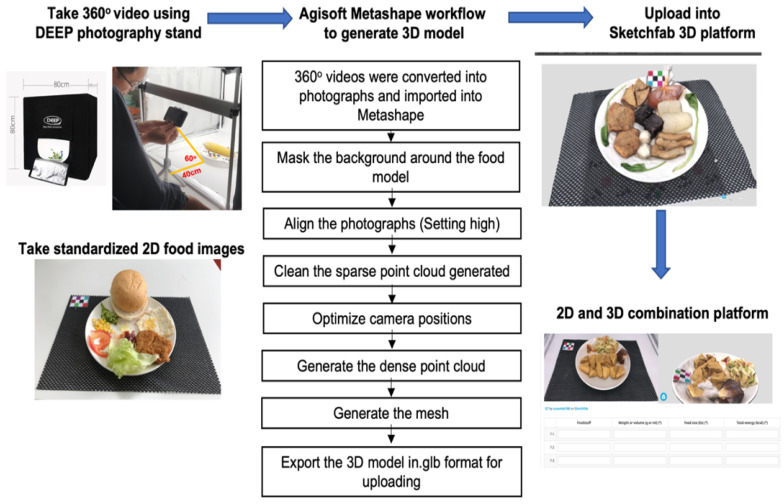
Study flowchart of educational integration of digital food models into a nutritional practicum course.

**Figure 3 nutrients-14-03313-f003:**
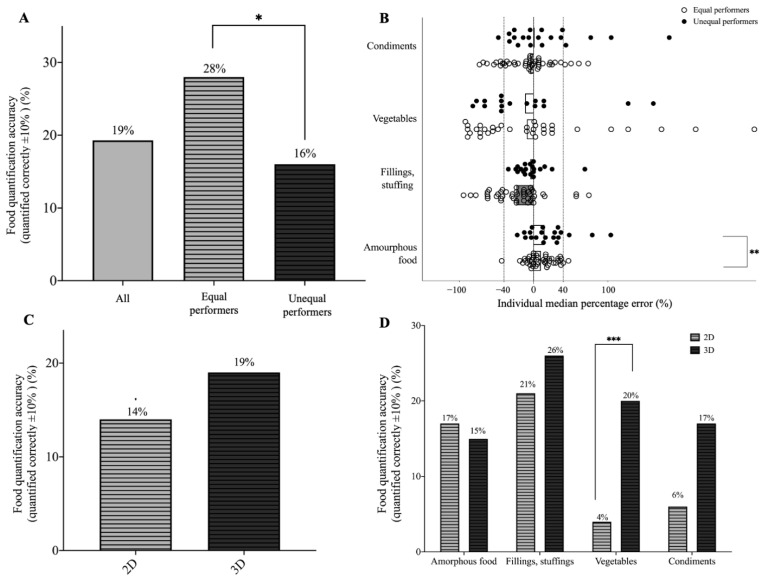
Accuracy of food quantification (**A**) and median percentage error (**B**) of estimated calories between equal (*n* = 45) and unequal (*n* = 20) performers in the first semester. Overall food quantification accuracies (within ±10% difference of total calories) (**C**) and stratified by food groups (**D**) between the 2-D and 3-D food models among unequal performers. Equal and unequal performers were defined using their performance difference between 2-D and 3-D estimates of <6 or ≥6 out of 12 tested food items, respectively, in the first semester. * *p* < 0.05; *** *p* < 0.001.

**Figure 4 nutrients-14-03313-f004:**
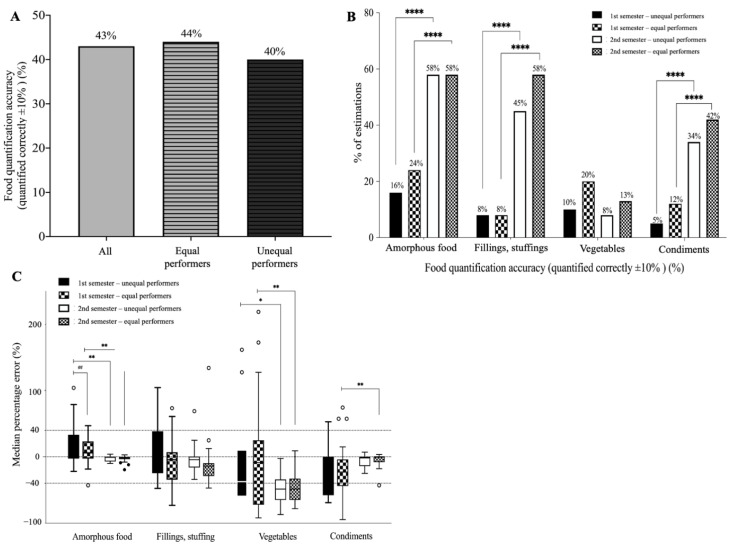
Overall food quantification accuracy (within ±10% difference of total calories) (**A**), estimation accuracy stratified by food groups (**B**), and median percentage error (**C**) of estimated calories in the second semester (*n* = 65). Equal and unequal performers were defined as a performance difference between 2-D and 3-D estimates of <6 or ≥6 out of 12 tested food items, respectively, in the first semester. ## *p* < 0.01 for comparing two groups of students; * *p* < 0.05; ** *p* < 0.01; **** *p* < 0.0001 for difference within groups over time.

**Table 1 nutrients-14-03313-t001:** Students’ performance of food identification and quantification of two-dimensional (2-D) and 3-D digital food models in the first semester (*n* = 65).

Food Set	2-D					3-D					Chi-Squared Test ^g^	Spearman Correlation ^f^
	Identified Correctly (%)	Quantified Correctly ±10% (%)	Over-	Under-	Omitted (%)	Identified Correctly (%) ^a^	Quantified Correctly ±10% (%) ^b^	Over-	Under-	Omitted (%) ^e^		*r; p* Value ^g^
			Estimated (%)	Estimated (%)				Estimated (%) ^c^	Estimated (%) ^d^			
Sweet corn	100.0%	21%	78%	1%	0%	100.0%	18%	81%	2%	0%	0.535	0.895; *p* < 0.0001
Sweet potato	100.0%	6%	45%	49%	0%	99.0%	6%	48%	46%	0%	NA	0.901; *p* < 0.0001
Red beans	99.0%	21%	25%	54%	0%	100.0%	24%	24%	52%	0%	0.508	0.779; *p* < 0.0001
Sugar	81.0%	12%	10%	60%	18%	85.0%	15%	9%	60%	16%	0.563	0.692; *p* < 0.0001
Red bean cake	99.0%	24%	46%	28%	1%	55.0%	19%	43%	36%	2%	0.410	0.704; *p* < 0.0001
Wonton	87.0%	33%	40%	12%	15%	88.0%	31%	39%	15%	15%	0.752	0.796; *p* < 0.0001
Noodles	97.0%	25%	36%	37%	1%	99.0%	30%	37%	31%	1%	0.453	0.782; *p* < 0.0001
Pork stuffing	90.0%	22%	42%	25%	10%	88.0%	24%	46%	19%	12%	0.766	0.745; *p* < 0.0001
Chicken leg	95.0%	19%	45%	31%	5%	88.0%	16%	48%	31%	4%	0.575	0.782; *p* < 0.0001
Oil	57.0%	36%	16%	1%	46%	57.0%	30%	22%	1%	46%	0.236	0.937; *p* < 0.0001
Sauce	81.0%	4%	15%	51%	30%	85.0%	3%	22%	45%	30%	0.436	0.795; *p* < 0.0001
Vegetables	73.1%	9%	24%	42%	25%	61.9%	15%	22%	37%	15%	0.181	0.876; *p* < 0.0001
Overall	89.2%	19.4%	35.2%	32.7%	12.7%	84.9%	19.3%	36.8%	31.3%	11.8%	0.968	

^a^ The proportion of students who correctly identified food items. ^b^ The proportion of students who quantified food calories within ±10% of ground truth calories. ^c^ Overestimated: The proportion of students who quantified food calories >10% of the ground truth total kcal. ^d^ Underestimated: The proportion of students who quantified food calories <−10% of the ground truth total kcal. ^e^ Omitted: students who failed to identify and quantify food items from images. ^f^ Spearman correlation for agreement between 2-D and 3-D. ^g^ Pairwise Chi-squared test comparing food accuracies between 2-D and 3-D images.

**Table 2 nutrients-14-03313-t002:** Students’ receptiveness and responses to the educational integration of digital food models for portion size education (*n* = 65).

	All (*n* = 65)	Equal Performers (*n* = 45)	Unequal Performers (*n* = 20)	*p* Value ^a^
**1. Which method was easier to identify food items?**				
Real food	31% (20/65)	27% (12/45)	40% (8/20)	0.383
2-D food image	23% (15/65)	22% (10/45)	25% (5/20)	0.99
Interactive 3-D food model	5% (3/65)	7% (3/45)	0% (0/20)	0.547
No difference between 2-D and 3-D	28% (18/65)	27% (12/45)	30% (6/20)	0.773
2-D and 3-D combination	14% (9/65)	18% (8/45)	5% (1/20)	0.255
**2. Which method was easier to quantify food items?**				
Real food	38% (25/65)	38% (17/45)	40% (8/20)	0.987
2-D food image	11% (7/65)	13% (6/45)	5% (1/20)	0.423
Interactive 3-D food model	3% (2/65)	4% (2/45)	0% (0/20)	0.9
No difference between 2-D and 3-D	18% (12/65)	18% (8/45)	20% (4/20)	0.921
2-D and 3-D combination	26% (17/45)	22% (10/45)	35% (7/20)	0.361
**3. Which approach was helpful to conduct virtual “dietary assessment” training?**				
2-D food image	15% (10/65)	18% (8/45)	10% (2/20)	0.711
Interactive 3-D food model	3% (2/65)	2% (1/45)	5% (1/20)	0.524
2-D and 3-D combination	71% (46/65)	67% (30/45)	80% (16/20)	0.413
None of them	11% (7/65)	13% (6/45)	5% (1/20)	0.788
**4. Do you think 2-D or 3-D training should be retained in the classroom in the future?**				
2-D food image	12% (8/65)	13% (6/45)	10% (2/20)	0.711
Interactive 3-D food model	3% (2/65)	4% (2/45)	0% (0/20)	0.988
2-D and 3-D combination	82% (53/65)	78% (35/45)	90% (18/20)	0.768
None of them	3% (2/65)	4% (2/45)	0% (0/20)	0.988

^a^*p*-value for Chi-square test comparing equal performers and unequal performers.

**Table 3 nutrients-14-03313-t003:** Students’ overall digital food viewing skills using a combination of two-dimensional (2-D) and interactive 3-D food models in the second semester (*n* = 65).

Food Item	Number of Estimates	Median [IQR] Percentage Error (%)	Food Identification Accuracy (%) ^a^	Quantified Correctly ±10% (%) ^b^	Over-Estimated (%) ^c^	Under-Estimated (%) ^d^	Omitted ^e^
Sweet potato	58	−9 [−25; 16]	74%	43%	24%	24%	10%
Rice	195	0 [−2; 0]	100%	95%	1%	5%	0%
Noodles	65	0 [−17; 4]	100%	63%	9%	28%	0%
Burger	65	4 [−3; 29]	100%	51%	35%	14%	0%
Dumplings	65	−53 [−68; −43]	100%	6%	2%	92%	0%
Bun	65	0 [−3; 0]	100%	9%	2%	89%	0%
Tempura	65	−27 [−37; −12]	100%	18%	8%	74%	0%
Chicken	65	0 [−3; 0]	100%	86%	5%	9%	0%
Pork stuffing	130	−6 [−29; 0]	100%	54%	23%	23%	0%
Fish	65	0 [−2; 3]	100%	75%	6%	18%	0%
Beef	65	−6 [−17; −0.5]	100%	71%	2%	28%	0%
Egg	130	0 [−5; 18]	100%	85%	15%	0%	0%
Oysters	58	−21 [−21; 18]	92%	21%	22%	52%	5%
Tofu	50	7 [−3; 46]	79%	50%	32%	8%	5%
Vegetables	202	−50 [−69; −27]	77%	11%	4%	61%	23%
Sauce	58	16.5 [−58; 150]	91%	26%	36%	30%	8%
Mayonnaise	44	50 [37; 71]	68%	0%	0%	71%	29%
Coating	46	−31 [−47; −31]	72%	0%	74%	5%	23%
Oil	366	0 [−14; 0]	85%	51%	10%	29%	10%
Overall			91.50%	42.90%	16.30%	35.00%	6.0%

^a^ The proportion of students who correctly identified food items. ^b^ The proportion of students who quantified food calories within ±10% of ground truth calories. ^c^ Overestimated: The proportion of students who quantified food calories >10% of the ground truth total kcal. ^d^ Underestimated: The proportion of students who quantified food calories < −10% of the ground truth total kcal. ^e^ Omitted: students who failed to identify and quantify food items from images.

## Data Availability

The datasets used and/or analyzed during the current study are available from the corresponding author on reasonable request.

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
