# Peer review of "COVID-19 and Virtual Nutrition: A Pilot Study of Integrating Digital Food Models for Interactive Portion Size Education"

_nutrients, 2022, doi:10.3390/nu14163313_

Round 1
Reviewer 1 Report
Dear authors, the manuscript is interesting and needs minor revisions. Please structure the abstract better. Please use Mesh Terms as keywords if possible.
The introductions and methodology are adequate.
It would be appropriate to better describe the sampling of participants and whether there are significant differences within.
Please make the results easier to read.
The discussions are trivial and should be expanded by explaining how it can evolve and any repercussions it could have.
Please check your English.
Kind regards.
Author Response
- Dear authors, the manuscript is interesting and needs minor revisions. Please structure the abstract better. Please use Mesh Terms as keywords if possible.
Response: Thank you for your positive comment and we sincerely appreciate your valuable comments and suggests. We already structure the abstract accordingly to your suggestion:
“Abstract: (page 1-2, line 25-49)
Background and aims: Digital food viewing are vital skills connecting dieticians to e-health. The aim of this study was to integrate a novel pedagogical framework which combines interactive three- (3D) and two-dimensional (2D) food models into a formal dietetic training course. The level of agreement between the digital food models (first semester) and the effectiveness of educational integration of digital food models during the school closure due to coronavirus disease 2019 (COVID-19) (second semester) were evaluated.
Method: In total, 65 second-year undergraduate dietetic students were enrolled in a nutritional practicum course in the School of Nutrition and Health Sciences, Taipei Medical University (Taipei, Taiwan). A 3D food model was created using Agisoft Metashape. Students’ digital food viewing skills and receptiveness towards integrating digital food models were evaluated.
Results: In the first semester, no statistical differences were observed between 2D and 3D food viewing skills in food identification (2D: 89% vs. 3D: 85%) and quantification (within ±10% difference in total calories) (2D: 19.4% vs. 3D: 19.3%). A Spearman correlation analysis showed moderate to strong correlations of estimated total calorie (0.69~0.93; all p values <0.05) between the 3D and 2D models. Further analysis showed that students who struggled to master both 2D and 3D food-viewing skills had lower estimation accuracies than those who did not (equal performers: 28% vs. unequal performers:16%, p=0.041), and interactive 3D models may help them perform better than 2D models. In the second semester, the digital food-viewing skills significantly improved (food identification: 91.5% and quantification: 42.9%) even for those students who struggled to perform digital food-viewing skills equally in the first semester (equal performers: 44% vs. unequal performers: 40%).
Conclusion: Although repeated training greatly enhanced students’ digital food-viewing skills, a tailored training program may be needed to master 2D and 3D digital food-viewing skills. Future study is needed to evaluate the effectiveness of digital food models for future “eHealth” care.”
We also tried to modify the keywords according to your suggestion:
Keywords: image-based dietary assessment; tele-dietetics; nutrition education; augmented and virtual reali-ty, distance education and online learning
- The introductions and methodology are adequate.It would be appropriate to better describe the sampling of participants and whether there are significant differences within.
Response: Thank you for your suggestions. Because our pedagogical framework was developed and tailored for the Nutritional Practicum (NP) course. The NP course is a kitchen-based laboratory course for second-year dietetic students which teaches practical skills involved in dietary assessments such as culinary skills, food classification, and food portion size estimation, we only targeted and included second year students who enrolled in NP course in the School of Nutrition and Health Sciences, Taipei Medical University (TMU) (Taipei, Taiwan) between study period. All enrolled students were informed about the research purposes in advance via email, and none declined to participate in this study. Therefore, in the context of our study, we couldn’t conduct sample size calculation and power analysis to calculate the sampling. We have added more details in the Method section:
“2. Materials and Methods (page 1-2, line 25-49)
2.1. Participants
Because our pedagogical framework was developed and tailored for the Nutritional Practicum (NP) course, the inclusion criteria was second year students enrolled in NP course in the School of Nutrition and Health Sciences, Taipei Medical University (TMU) (Taipei, Taiwan) between September 2019 and July 2020. All enrolled students were informed about the research purposes in advance via email, and none declined to participate in this study. The study protocol was approved by the Institutional Review Board of TMU (N201904035).”
We also acknowledge the limitation of sampling in the Discussion section:
“Discussion ((page 1-2, line 25-49)
Another limitation is the participants sampling. As the study took the second-year dietetic students in NP course as the target learner, we could not provide the sample size cal-culation and power analysis in the context of this study. Limited numbers of food items and participants also highlight the need for future collaborations in order to recruit a large-scale representative population and confirm the validity and effectiveness of the new digital food portion size educational paradigm.”
- Please make the results easier to read. The discussions are trivial and should be expanded by explaining how it can evolve and any repercussions it could have.
Response: Thank you for your valuable comments and suggests. We already added in to the discussion accordingly to your suggestion:
“Discussion (page 15, line 94-109)
Food image viewing is a vital skill that connects dieticians to e-health. One of the greatest advantages of conducting food portion size training via 2D and 3D model is the ability for students to gain access to the material from anywhere and the functionality of being able to move/magnify. Our framework can be used to replace the traditional face to face dietetic training and boost students' motivation and adherence to virtual e-learning in the era of COVID-19. Additionally, the MAR-3D environment allows students to view multiple models simultaneously and compare them to each other, which is more difficult to do with 2D tools. Virtual and augmented reality, however, present the cutting-edge technology that can depict realistic food 3D models and concepts to students. While the quality of the 3D models can be improved with the use of better 3D scanners, photogrammetry software and algorithm, the level of detail of food that can be represented in the MAR viewpoint is limited. Our study is the first step in the establishment of a more complete research program for evaluating dietetic training curriculum and future development of integrating new technology into portion size education. Therefore, we propose to utilize the digital food models that allow students to observe and manipulate them from VR and AR viewpoints in the following study
Reviewer 2 Report
Thank you for the opportunity to review your manuscript. The paper seems interesting and presenting a logical arrangement of the content, although I suggest that the Authors think through some revisions:
- the phrase 'pilot study' should be included in the title of the manuscript.
- you relate the paper to the COVID-19 pandemic, and in the introduction the description of the background of the problem is short and in my opinion insufficient, please expand the question of how the pandemic affected education.
- please state the main objective and to it the research questions or hypotheses, instead of the several objectives present in the text.
- What were the criteria for qualifying participants and whether the group was representative - please justify in the methodology.
- please add in the conclusions more information on the practical application of the described method (use in specific types of school/university activities, in dietary or psychological counseling, in industry, etc.).
Congratulations on a great job! Greetings!
